# Bond Strength of Metallic or Ceramic Orthodontic Brackets to Enamel, Acrylic, or Porcelain Surfaces

**DOI:** 10.3390/ma13225197

**Published:** 2020-11-17

**Authors:** Mónica Pinho, Maria C. Manso, Ricardo Faria Almeida, Conchita Martin, Óscar Carvalho, Bruno Henriques, Filipe Silva, Afonso Pinhão Ferreira, Júlio C. M. Souza

**Affiliations:** 1Faculty of Health Sciences (FCS), Universidade Fernando Pessoa (UFP), 4249-004 Porto, Portugal; monicap@ufp.edu.pt (M.P.); cmanso@ufp.edu.pt (M.C.M.); 2Fernando Pessoa Energy, Environment and Health Research Unit (FP‑ENAS), University Fernando Pessoa (UFP), 4200-150 Porto, Portugal; 3LAQV, REQUIMTE, University of Porto (UP), 4050-313 Porto, Portugal; 4School of Dentistry (FMDUP), University of Porto (UP), 4200-135 Porto, Portugal; rfaperio@gmail.com (R.F.A.); apinhaoferreira@gmail.com (A.P.F.); 5School of Dentistry, University Complutense of Madrid (UCM), 28040 Madrid, Spain; conchitamartin@odon.ucm.es; 6Center for MicroElectromechanical Systems (CMEMS-UMINHO), University of Minho, 4800-058 Guimarães, Portugal; Oscar.carvalho@gmail.com (Ó.C.); brunohenriques@dem.uminho.pt (B.H.); fsamuel@dem.uminho.pt (F.S.); 7Department of Mechanical Engineering (EMC), Federal University of Santa Catarina (UFSC), Florianópolis 88040-090, Brazil; 8Department of Dental Sciences, University Institute of Health Sciences (IUCS), CESPU, 4585-116 Gandra PRD, Portugal

**Keywords:** shear bond strength, tensile bond strength, orthodontic brackets, adhesion

## Abstract

Bonding strategies within different brackets and dental materials are still a challenge concerning adhesion and dental surface damage. This study compared the shear and tensile bond strength of orthodontic ceramic and metallic brackets to enamel, acrylic, and ceramic surfaces after thermal cycling. Dental surfaces were divided into three groups: enamel, ceramic, and acrylic. Each group received stainless-steel and ceramic brackets. After thermal cycling, specimens were randomly divided into two subgroups considering tensile (TBS) or shear bond strength (SBS) test. After the mechanical testing, scanning electron and optical microscopy were performed, and the adhesive remnant index (ARI) was determined. The two-way ANOVA full factorial design was used to compare TBS, SBS, and ARI on the surface and bracket type (α = 0.05). There were significant differences in TBS, SBS, and ARI values per surface (*p* < 0.001 and *p* = 0.009) and type of bracket (*p* = 0.025 and *p* = 0.001). The highest mean SBS values were recorded for a ceramic bracket bonded to an acrylic surface (8.4 ± 2.3 MPa). For TBS, a ceramic bracket bonded to acrylic showed the worst performance (5.2 ± 1.8 MPa) and the highest values were found on a metallic bracket bonded to enamel. The adhesion of metallic or ceramic brackets is enough for clinical practice although the damage of the enamel surface after debonding is irreversible and harmful for the aesthetic outcome of the teeth.

## 1. Introduction

The increasing demand for orthodontic treatment keeps orthodontists chasing the optimal bonding strategy considering different brackets and surfaces [1,2,3]. Since most adult patients have dental restorations and/or prostheses, the bond strength of brackets to enamel and restorative material is a major concern during orthodontic treatment and later at the bracket removal [4]. Another concern is that the removal of the bracket may result in irreversible damage to the restorative and enamel surfaces [5,6]. Currently, all-ceramic restorations are widely used to restore missing or damaged enamel, and therefore several types of ceramics have been developed during recent years. The most common ceramic-based materials for veneering prosthetic structures include feldspar and leucite-based porcelain [7,8]. Such materials can be manufactured by traditional laboratory methods or by CAD-CAM [9]. CAD-CAM approach leads to a less time-consuming technique with lower sensitivity and therefore the fitting accuracy of the prosthetic is more predictable when compared to traditional laboratory methods (e.g., hot-pressing technique) [10,11]. Considering ceramic materials possess high chemical resistance, the etching of surfaces must be performed using hydrofluoric acid prior to bracket adhesion. Acidic etching results in a porous layer, allowing the penetration of low viscosity resin-based adhesives, that lead to a mechanical interlocking after the adhesive polymerization [12,13,14]. In addition, chemical bonding can be achieved by applying a silane coupling prior to the resinous adhesive [15,16].

Interim crowns are often required to re-establish aesthetics, function and soft tissues apposition while the final ceramic-based restoration is manufactured, thus working as a pattern for the final restoration [17,18]. Self-curing polymers based on acrylic or bis-acrylic are well-known materials for manufacturing interim crowns [19]. Adhesive bonding agents for acrylic materials are still scarce leading to the use of standard techniques for enamel bonding. That may not match the adhesion requirements, due to the chemical composition and roughness of the substrate [20,21]. Indeed, the acidic etching used for acrylic surfaces does not increase the roughness, although it can remove residues which negatively affect the bracket adhesion [18]. A previous study suggested the use of adhesives with the same chemical composition of the provisional crowns [22]. However, other studies suggested performing the conventional etching followed by the adhesive application [18,23,24].

The bond strength at the bracket–adhesive–substrate interface must withstand forces during orthodontic treatment, although it should also allow the removal of the brackets without fractures of those substrates, namely restorative materials or tooth enamel [25]. In fact, novel developments have been seeking an efficient and safe method for debonding brackets by using a wide variety of tools and procedures [6,26,27,28]. The detachment must occur at the bracket–adhesive interface to prevent any damage of dental surfaces [26,29,30]. Thus, the main aim of this study was to compare the shear and tensile bond strength of ceramic or metallic orthodontic brackets to enamel, acrylic, and ceramic surfaces after thermal cycling. The null hypothesis of this study was that there are no differences in shear and tensile bond strength between brackets bonded to enamel, acrylic, and ceramic surfaces.

## 2. Materials and Methods

### 2.1. Preparation of Brackets for Dental Substrate Samples

A total of 132 specimens were classified into three major groups: group I: composed of 44 human teeth; group II: consisting of 44 acrylic crowns (poly-methyl-methacrylate, Dentalon Plus, Kulzer GmbH, Hanau, Germany); and group III: comprising 44 crowns of leucite-based glass–ceramics (IPS Empress system, Ivoclar Vivadent, Liechtestein). Each tooth or crown received two brackets, a stainless-steel bracket (MB, Master Series, American Orthodontics, Sheboygan, WI, USA) and a ceramic bracket (CB, Radiance Brackets, Empower Clear Brackets, American Orthodontics, Sheboygan, WI, USA). Surfaces were then randomly divided into two subgroups, according to the tests to which they were submitted: subgroup 1 included the brackets subjected to shear bond strength test; and subgroup 2 included the brackets subjected to tensile bond strength test.

Prior to bonding, teeth with undetermined times of extraction were cleaned from any residual soft tissues and then immersed in chloramine solution at 4 °C over a period of 7 days. Then, teeth were stored in distilled water at a temperature of 4 °C for 7 days for maintenance of the chemical stability and hydration of the teeth tissues [31,32,33]. After that, the teeth were cleaned with water and brushed at low speed and then rinsed with water spray and air-dried under oil-free airstream for 3 s. The bonding surface was etched using 37% phosphoric acid (Octacid, Clarben S.A., pH < 2 at 20 °C) for 30 s followed by thorough washing for 60 s and gently drying under airstream for 3 s. The ceramic crowns were polished with Arkansas stone and then conditioned with 9.6% hydrofluoric acid (Porcelain Etch Gel, Pulpdent, pH < 1.5 at 20 °C) for 30 s followed by thorough washing and drying (same protocol described before for teeth enamel). At first, acrylic surfaces were polished with Arkansas stone and then conditioned with 37% phosphoric acid for 30 s followed by thorough washing and drying (same protocol described before for teeth enamel).

The inner middle region of the bracket coated with the adhesive was pressed firmly against the enamel and the adhesive was removed around the bracket. Transbond XT™ adhesive (3M Unitek, Monrovia, CA, USA) was light cured by using a LED light cure unit for 20 s, as recommended by the manufacturer. One operator carried out the bonding procedure of brackets to dental surfaces for all groups by using Transbond XT™ adhesive system.

### 2.2. Thermal Cycling Tests

After adhesive procedures, all specimens were thermal cycled in Fusayama’s artificial saliva at pH 5.5 [31,32,33]. The corrosive and lubricated effect of the artificial saliva solution used in this study has been reported to be similar as in human saliva [31,32,33]. The chemical composition of the artificial saliva is shown in Table 1. Each specimen underwent 4000 complete cycles in artificial saliva between 5 and 55 °C, according to the ISO/TS 11405-2003, [31] with a dwell time of 15 s for each bath and a transfer time between baths of 25 s, over a period of 45 s [32,33].

### 2.3. Shear and Tensile Bond Strength Tests

For the strength tests, each specimen was embedded in Epoxy (EpoFix™, Izasa, Portugal) acrylic resin after thermal cycling and stored in distilled water at room temperature until the tests were carried out, to avoid dehydration of the samples. Each specimen was then loaded onto a universal test machine (Instron, Norwood, MA, USA) with a load cell of 25 kN. The shear and tensile bond strength tests were performed according to the ISO/TS 11405-2003 guidelines [31]. The shear force was applied parallel to the interface of the bonding surfaces and the tensile force perpendicular to the interface at a speed of 0.5 mm/min until bonding failed. Forces were recorded automatically at the point of failure in N and converted into MPa for purposes of better comparability, in accordance with the following equation:*R* (N/mm^2^) = *F* (N)*B* (mm^2^)(1)
where *R* stands for bond strength, *F* for force and *B* is the bracket area.

### 2.4. Morphological Analyses of Surfaces and Interfaces

Samples of each group, subjected or not to thermal cycling, were randomly chosen for scanning electron microscopy (SEM) observation of the enamel–adhesive–bracket region. The samples were mounted in acrylic resin and cross-sectioned perpendicularly to the bracket–adhesive interface plan. The cross-sectioned specimens were wet ground on silicon carbide papers down to 2500 mesh and polished using a 1 μm diamond slurry before SEM inspection. SEM analyses were performed on secondary (SE) and backscattered (BSE) electron mode, at a magnification ranging from ×100 to ×2000 at 15 kV. Surfaces were previously sputter-coated with a AuPd film. The brackets of each group were also analyzed by energy dispersive spectroscopy (EDS) after the thermal cycling and the strength tests. After debonding, each specimen was photographed at a magnification of ×50 using a DM2700 M Leica^TM^ optical microscope (Leica Camera Inc., Allendale, NJ, USA) and the remaining adhesive area was determined using ImageJ 2.0.0 (ImageJ2 Research Services Branch, NIH, Bethesda, MD USA).

### 2.5. Statistical Analysis

Data were analyzed using the IBM SPSS Statistics 22.0 software (IBM Corp., Armonk, NY, USA), considering a significance level of 0.05 (α = 0.05). Data on shear bond strength (SBS) and tensile bond strength (TBS) values (MPa) and the adhesive remnant index (ARI) were described based on their mean and corresponding standard deviation, as well as minimum and maximum, for each surface (enamel, ceramic and acrylic) and bracket type (stainless-steel and ceramic). The correlation between ARI and the SBS or TBS values was also studied. Further comparison of strength values (MPa) and ARI by surface type and bracket type was done using a two-way ANOVA full factorial design. In most groups, the data for both variables proved to be approximately normally distributed (Shapiro–Wilk test) and when non-normal distributions were attained these were not caused by symmetry issues. Homogeneity of variances was obtained (Levene test). Upon detection of significant differences regarding surface material, these were analyzed using the post-hoc Tukey test, and its association with the bracket type was studied by a *t*-test.

## 3. Results

Results revealed different bond strength values for the surfaces. The comparison of SBS and TBS between different dental surfaces and types of bracket can be seen in Table 2.

The two-way ANOVA detected significant differences in SBS and TBS values per surface (*p* < 0.001 and *p* = 0.009, respectively) and on the type of bracket (*p* = 0.025 and *p* = 0.001, respectively). Regarding SBS, significant differences were found only for the (CB) ceramic brackets in acrylic surfaces, which showed the highest mean values (8.4 ± 2.3 MPa), followed by the leucite glass–ceramic surfaces (6.8 ± 2.3 MPa) and tooth enamel (4.7 ± 1.5 MPa). TBS values showed significant differences only between acrylic (6.4 ± 3.2 MPa) and enamel (8.9 ± 3.1 MPa) with a (MB) stainless-steel bracket. No significant interaction was detected between the surface (*p* = 0.140) and the bracket type (*p* = 0.666) (Table 2).

In the stainless-steel bracket group, there was no significant difference in SBS between the three surfaces (*p* = 0.339). On the other hand, there were significant differences in the TBS values (*p* = 0.049), as they were significantly higher in enamel (8.9 ± 3.1 MPa) and significantly lower in acrylic (6.4 ± 3.2 MPa). The TBS of ceramic brackets did not differ significantly between enamel (*p* = 0.279) and acrylic (*p* = 0.6). The SBS of ceramic brackets showed significant differences between the three surfaces (*p* < 0.001), with the lowest strength values found in enamel (4.7 ± 1.5 MPa) and the highest in acrylic surfaces (8.4 ± 2.3 MPa). There was no significant difference in TBS between the three surfaces (*p* = 0.172).

Significant differences were found between bracket types regarding enamel, as those with stainless-steel brackets (SBS, 6.9 ± 3.2 MPa; TBS, 8.9 ± 3.1 MPa) showed better results than those with ceramic brackets (SBS, 4.7 ± 1.5 MPa; TBS, 6.6 ± 2.8 MPa).

The results of the ARI comparison using a two-way ANOVA, on the surface material and bracket type, are shown in Table 3. Regarding SBS, significant differences were found in the mean values of ARI for surface material (*p* = 0.027) and bracket type (*p* < 0.001) but no significant interaction was found between surface material and bracket type (*p* = 0.125) (Table 3).

The mean values of ARI regarding TBS showed significant differences only for surface material (*p* < 0.001), and not for bracket type (*p* = 0.158) nor for the interaction between surface material and bracket type (*p* = 0.597). The mean values of both SBS and TBS of stainless-steel brackets were significantly higher for glass–ceramic surfaces and lower for acrylic. However, no significant differences were detected regarding the other two teeth surfaces. Significant differences were found in SBS mean values between stainless-steel and ceramic brackets on enamel (15.3 ± 13.8 vs. 29.6 ± 11.2, *p* = 0.001) and on acrylic (9.2 ± 10.6 vs. 30.4 ± 10.9, *p* < 0.001), with ceramic brackets showing significantly higher mean SBS values (32.3 ± 19.7). Significant differences in mean TBS values were detected only for acrylic surfaces (21.8 ± 13.7, *p* = 0.014), with ceramic brackets showing significantly higher mean TBS values (35.8 ± 22.3).

SEM images revealed different morphological aspects of the surfaces after acid-etching treatment, as seen in Figure 1. That provided important information concerning the resulting topography.

Teeth enamel showed transversally oriented prisms with small holes before etching (Figure 1A) and, after 37% phosphoric acid etching for 30 s. The depth and morphology of the demineralized zones increased after removal of hydroxyapatite (Figure 1B). A similar pattern was noted in polymethyl-methacrylate surfaces before and after etching (Figure 1C,D). Analysis of the leucite glass–ceramic surface showed an irregular but not porous surface after glass phase removal and before etching (Figure 1E). An irregular porous surface was seen after hydrofluoric-acid etching on leucite glass–ceramic (Figure 1F), with large and deep voids and channels. Additionally, leucite crystals were easily noticed protruding from the glassy matrix.

SEM images of interfaces involving brackets, adhesive, and the human enamel, glass–ceramic, or acrylic surfaces before and after thermal cycling are shown in Figure 2, Figure 3, Figure 4 and Figure 5.

The thickness of the composite adhesive varied between specimens ranging from 70 to 200 μm (Figure 2, Figure 3, Figure 4 and Figure 5). However, there were significant differences in thickness between groups. The composite adhesive showed a high percentage of dispersed irregular particles ranging from 1 to 30 μm that were well distributed in the polymeric matrix (Figure 2, Figure 3, Figure 4 and Figure 5).

Structural defects and cracks were noted at the bracket–adhesive–substrate interfaces after thermal cycling (Figure 2, Figure 3, Figure 4 and Figure 5). Cracks were more noticeable between the ceramic bracket and the adhesive in the teeth group (Figure 3) and between the adhesive and acrylic surfaces (Figure 5). Leucite-based glass–ceramic specimens showed the fewest changes after thermal cycling (Figure 5).

After debonding in the teeth enamel group, EDS spectra of the bracket mesh base revealed the presence of chemical elements transferred from the enamel surface to the adhesive as seen in Figure 6. The amount of Ca embedded in the adhesive was around 21.84 ± 12.29%.

## 4. Discussion

The present study evaluated the (SBS) shear and (TBS) tensile bond strength of orthodontic ceramic and metallic brackets to enamel, acrylic, or glass–ceramic surfaces, after thermal cycling. In addition, the adhesive remnant index was studied by optical and electron scanning microscopy. The results rejected the null hypothesis of the present study, as they revealed the influence of the chemical composition and surface of the brackets on their tensile and shear bond strength to dental surfaces.

The type of bracket material was revealed to be an important factor for bond strength. The stainless-steel brackets showed higher bond strength values than that for ceramic brackets, for all groups. There is no consensus in the literature on which bracket type provides better adhesion [34,35,36,37,38,39,40]. However, the stainless-steel brackets used in this experiment had a mesh base, which is described in the literature as providing higher mechanical interlocking for the orthodontic adhesive [37], and, thus, those brackets were expected to provide higher bond strength. The only exception was found in the acrylic group, in which the ceramic brackets performed better than the stainless-steel brackets considering SBS test, but differences were not statistically significant. The teeth enamel group (*p* = 0.049) showed higher bond strength in the TBS test and lower bond strength in the SBS test when compared to the other groups. This finding can be related to the porous enamel as a result of the acid-etching, which may provide mechanical interlocking of the adhesive leading to an increase in tensile bonding.

In our study, some results showed lower bond strength values than those clinically recommended in the literature, since they should be higher than the threshold at 5–6 MPa for clinical use [39,41]. Although these results were also found in other previous studies [1,24,39,42,43,44,45,46,47,48], the findings in the present study can be related to the use of upper incisor brackets with a small base area and an anatomic shape not adjusted to the substrate shape. In fact, the upper incisor brackets were bonded to maxillary first molar teeth or prosthetic crowns (acrylic and glass–ceramic) with this same shape to avoid bias withing groups.

For both SBS and TBS, the mean values of ARI in this study showed a heterogeneous distribution for the three surfaces with percentages below 35.8 ± 22.3 in all groups. However, no surface cracks or fractures were noticed, meaning that the majority of the specimens could be included in the ARI level 1, as described by Artun and Bergland [49]. Still, no correlation was found between ARI and bond strength, which is in agreement with another previous study [50]. Other studies have used SEM-EDS for bracket adhesion evaluation [51,52,53,54] as performed in this study. The strategic method used in our study can indicate the presence of Ca on the adhesive fracture surfaces, that is related to the mechanical interlocking of the adhesive to the porous enamel. The main challenge after orthodontic treatment is to remove the orthodontic bracket and adhesive without damaging the enamel or restorative surface [55]. EDS results from this study demonstrated Ca deposits in the adhesive after debonding, even though no damaged surfaces were seen by microscopic examination. Thus, the enamel damage occurred after acid etching prior to the adhesion procedure and after debonding.

Results of in vitro tests provide important information for in vivo clinical practice, although different methods can be found in the literature [55]. Several factors influence the bond strength of brackets to dental surfaces, such as study design [45,56,57,58], bracket material [35,36,37,38,59,60], substrates [5,23], adhesive polymerization [43,61] and thermal cycling stimuli [57]. The acid-etched surfaces showed a very different pattern, and that seems to also influence the adhesion to the surfaces in our study. Moreover, that is in agreement with the literature, since not only the acid but also the etching time may affect such parameter [42,43,44,45,46,47,62,63].

A conventional acid-etching technique associated with a light-curing adhesive system composed of bis-GMA and TEGMA primer and with an adhesive paste on bis-GMA, silica, silane (SiH4), *n*-dimethyl-benzocaine, and hexafluorophosphate have been reported in the literature as the first choice for bonding brackets on enamel [42,53,63]. Therefore, the same adhesive system was used on the surfaces to promote standard bonding conditions in the present study [1,37,45,48,62,64,65,66]. The mode and time of light curing of the adhesive also influences the bond strength of brackets to the surfaces. High values of bond strength were found in the literature for orthodontic adhesives polymerized to ceramic or metallic brackets by using LED light-curing [37,67,68]. Moreover, the light source was reported to be a decisive factor for the bond strength of orthodontic brackets to other dental surfaces [43,61].

## 5. Conclusions

Within the limitations of an in vitro study, the main outcomes of the present study are the following:Surface and bracket type are important factors for bond strength since morphological aspects of the surface can vary depending on the acidic etching and chemical composition of the materials;Metallic brackets showed different a mesh design when compared to ceramic brackets, that can increase the mechanical interlocking of the adhesive;The size and anatomic adjustment of the bracket base to dental surface seems to be crucial for bonding. The misfit in shape between bracket and dental substrate alters the thickness of the adhesive, which affects the bond strength values;Shear and tensile bond strength does not significantly differ. There is no relationship between the number of adhesive remnants on the fracture surface and bond strength values.The acid etching procedure cause irreversible damage of the tooth enamel surfaces. Additionally, debonding of orthodontic brackets can increase the enamel damage even when that is not noticeable either clinically or under microscopic analyses. A balance in bond strength and enamel damage should be investigated in further studies.

## Figures and Tables

**Figure 1 materials-13-05197-f001:**
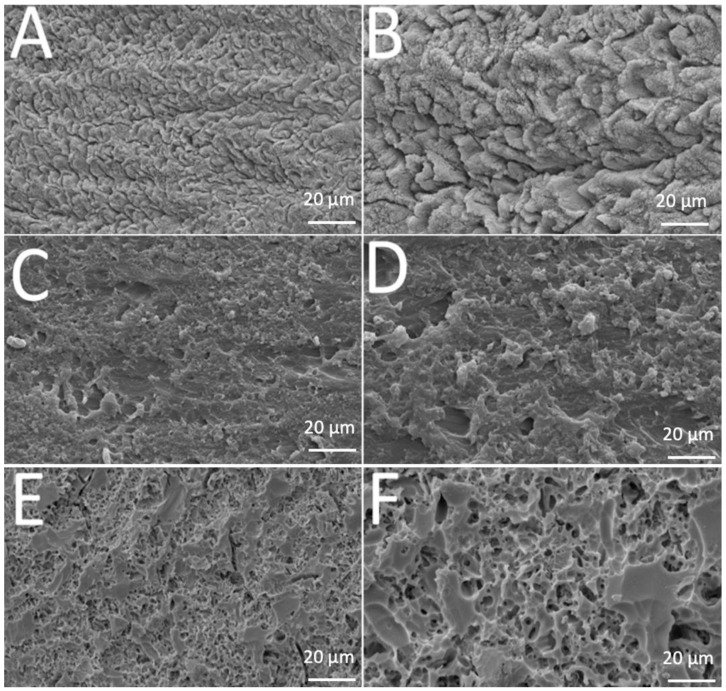
SEM images of (**A**,**B**) enamel, (**C**,**D**) acrylic or (**E**,**F**) leucite-based glass–ceramic after etching procedure.

**Figure 2 materials-13-05197-f002:**
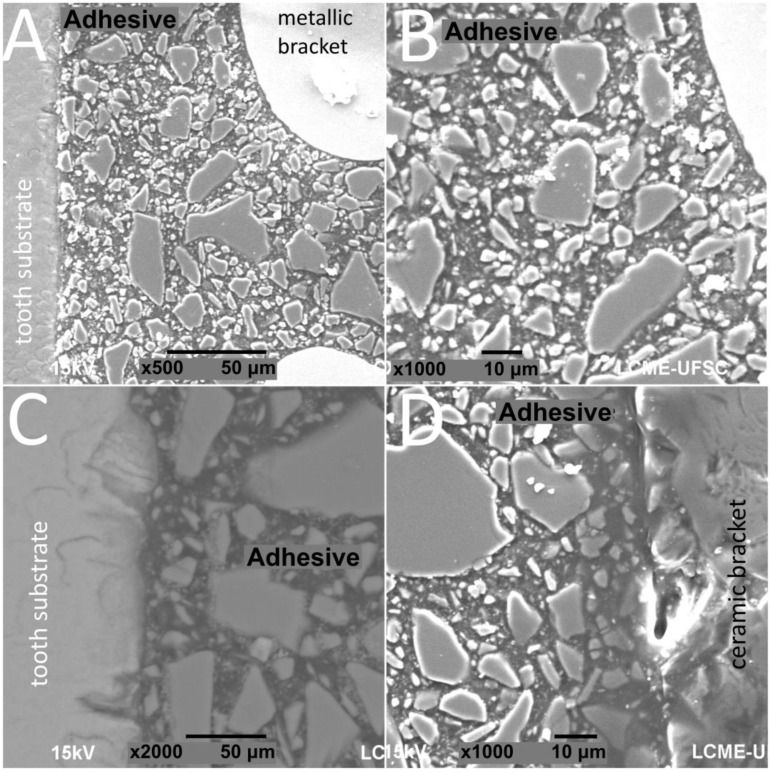
SEM images of cross-sectioned (**A**,**B**) metallic or (**C**,**D**) ceramic brackets bonded to enamel free of thermal cycling.

**Figure 3 materials-13-05197-f003:**
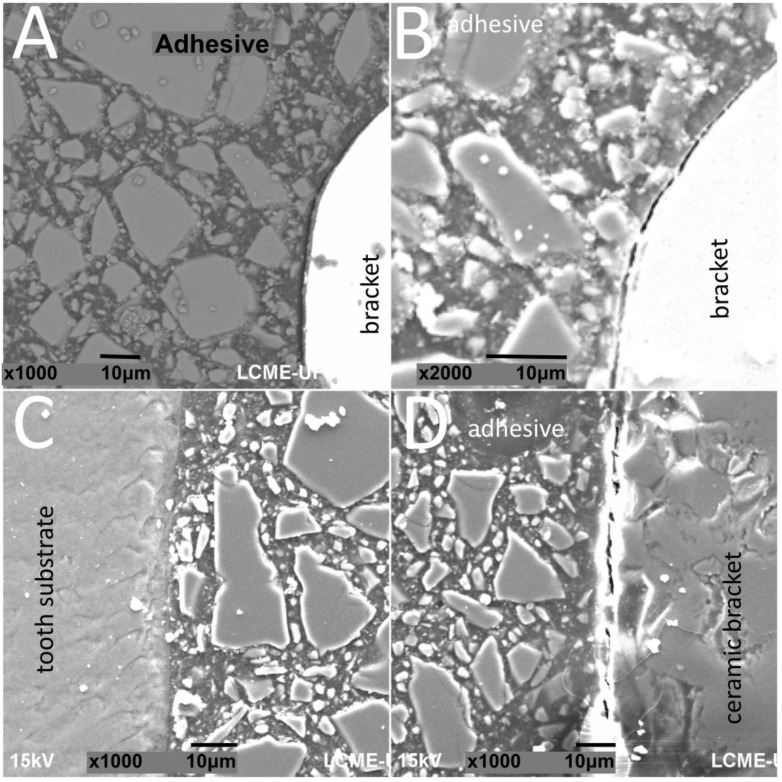
SEM images of cross-sectioned (**A**,**B**) metallic or (**C**,**D**) ceramic brackets bonded to enamel surfaces after thermal cycling.

**Figure 4 materials-13-05197-f004:**
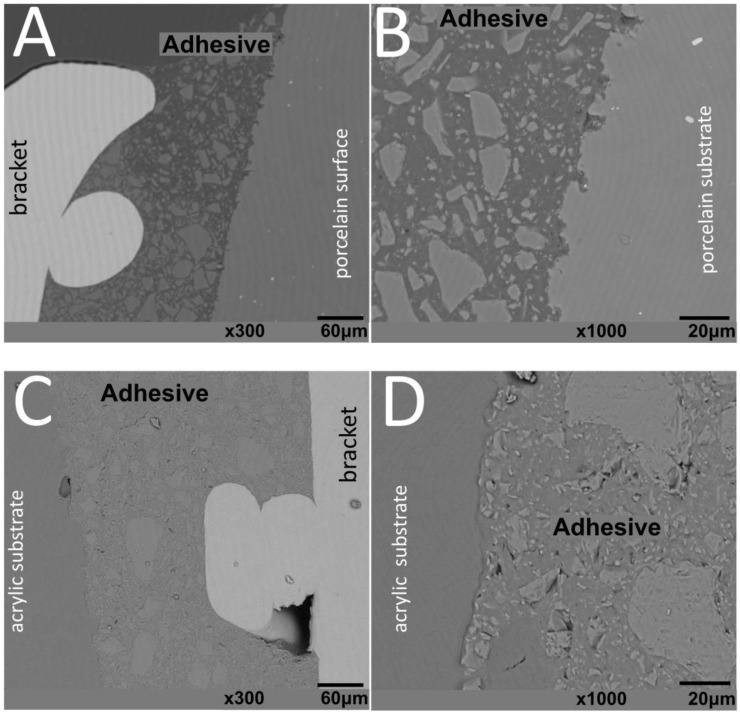
SEM images of metallic brackets bonded to (**A**,**B**) glass–ceramic or (**C**,**D**) acrylic surfaces.

**Figure 5 materials-13-05197-f005:**
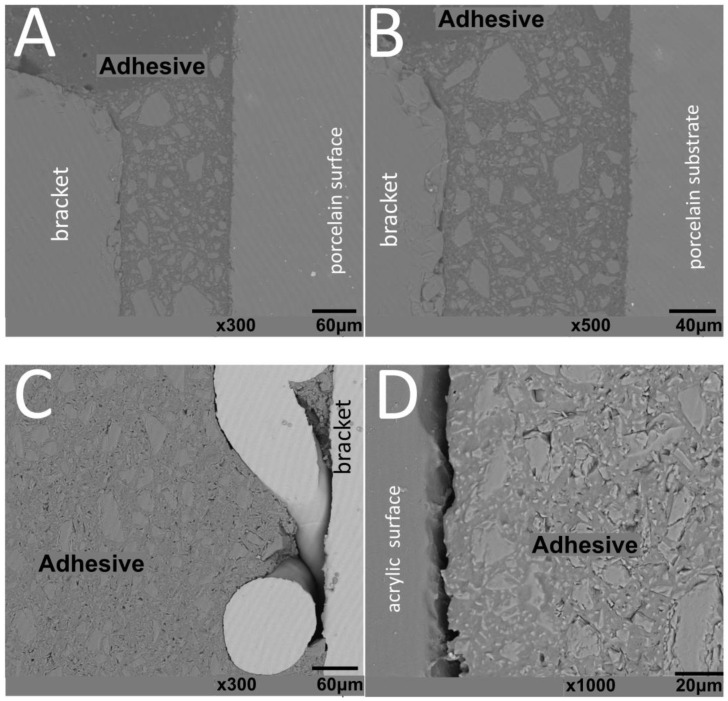
SEM images of ceramic brackets bonded to (**A**,**B**) glass–ceramic or (**C**,**D**) acrylic surfaces after thermal cycling.

**Figure 6 materials-13-05197-f006:**
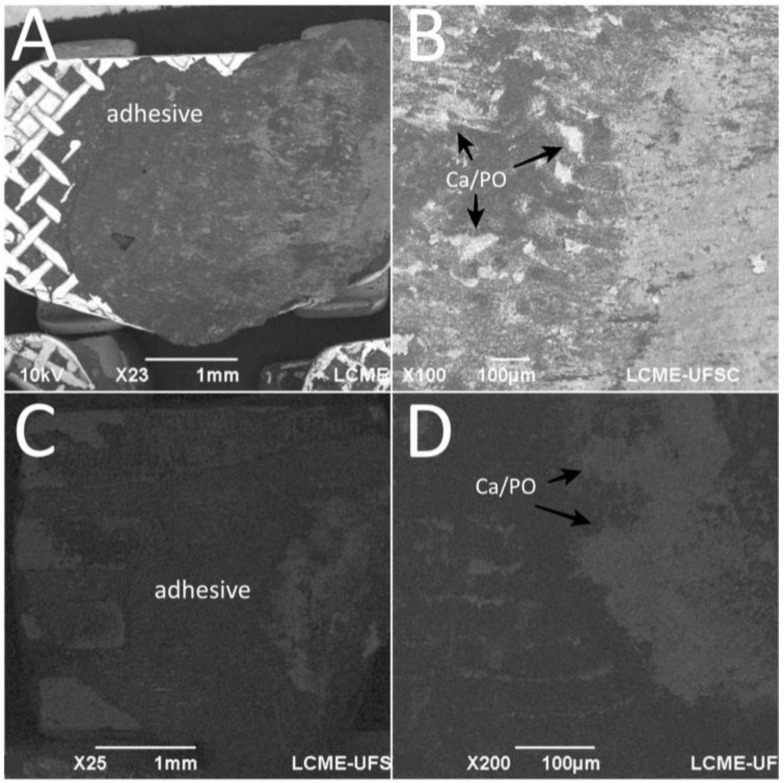
SEM images the bracket base after debonding from tooth enamel surfaces. EDS detected Ca and P embedded in the adhesive. Surface of the adhesive after debonding from (**A,B**) stainless-steel and (**C,D**) ceramic brackets.

**Table 1 materials-13-05197-t001:** Chemical composition of the Fusayama’s artificial saliva [31,32,33].

Compound	Content (g/L)
NaCl	0.4
KCl	0.4
CaCl_2_.2H_2_O	0.795
Na_2_S.9H_2_O	0.05
NaH_2_PO_4_.2H_2_O	0.69
Urea	1

**Table 2 materials-13-05197-t002:** Comparison of mean shear or tensile bond strength values (MPa) by a two-way ANOVA, surface and bracket type.

Surface		Shear Bond Strength		*p ***	Tensile Bond Strength		*p ***
		MB	CB		MB	CB	
**Enamel**	**n**	18	18	-	18	18	-
	**mean ± SD**	6.9 ^A^ ± 3.2	4.7 ^c,B^ ± 1.5	0.006	8.9 ^a,A^ ± 3.1	6.6 ^B^ ± 2.8	0.024
	**Min–max**	2.9–16.4	1.3–7.1	-	3.6–15	2.6–11.2	-
**Ceramic**	**n**	18	18	-	18	18	-
	**mean ± SD**	8.1 ± 3	6.8 ^b^ ± 2.3	0.102	7.4 ^a,b^ ± 2.9	5.7 ± 2.2	0.055
	**Min–max**	4.6–14.6	1.4–10.8	-	1.8–12.8	1.7–9	-
**Acrylic**	**n**	18	18	-	18	18	-
	**mean ± SD**	8.2 ± 3.7	8.4 ^a^ ± 2.3	0.841	6.4 ^b^ ± 3.2	5.2 ± 1.8	0.187
	**Min–max**	0.7–13.7	5.6–12.1	-	2.7–15.8	2.2–8.4	-
***p*** *	-	0.339	<0.001	-	0.049	0.172	-

(MB) stainless-steel bracket; (CB) ceramic bracket; (SD) standard deviation; ^a,b,c^ different letters stand for significant differences in mean values of force regarding surface material according to the * Tukey test. ^A,B^ different letters stand for significant differences in mean values of force regarding bracket type, according to the ** *t*-test.

**Table 3 materials-13-05197-t003:** Comparison of mean shear or tensile bond strength values (MPa) by a two-way ANOVA, surface and bracket type.

Surface		Shear Bond Strength			Tensile Bond Strength		
		MB	CB	*p ***	MB	CB	*p ***
**Enamel**	**n**	18	18	-	18	18	-
	**mean ± SD**	15.3 ^a,b,B^ ± 13.8	29.6 ^A^ ± 11.2	0.001	25.0 ^a,b^ ± 16.1	30.0 ± 18.3	0.380
	**Min–max**	0–41.6	6.8–51	-	1.1–59.3	4–58.8	-
**Ceramic**	**n**	18	18	-	18	18	-
	**mean ± SD**	23.8 ^a^ ± 15.2	32.3 ± 19.7	0.118	35.2 ^a^ ± 27	35.8 ± 22.3	0.939
	**Min–max**	0–60.4	1.9–73.3	-	0–100	7.5–80.6	-
**Acrylic**	**n**	18	18	-	18	18	-
	**mean ± SD**	9.2 ^b,B^ ± 10.6	30.4 ^A^ ± 10.9	<0.001	12.6 ^b,B^ ± 6.5	21.8 ^A^ ± 13.7	0.014
	**Min–max**	0–27.5	8.7–55.6	-	0–28.3	1.9–56.7	-
***p** **	-	0.004	0.834	-	0.002	0.073	-

(MB) stainless-steel bracket; (CB) ceramic bracket; (SD) standard deviation; ^a,b^ different letters stand for significant differences in mean values of force regarding surface material according to the * Tukey test. ^A,B^ different letters stand for significant differences in mean values of force regarding bracket type, according to the ** *t*-test.

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
