# Peer review of "Bond Strength of Metallic or Ceramic Orthodontic Brackets to Enamel, Acrylic, or Porcelain Surfaces"

_materials, 2020, doi:10.3390/ma13225197_

Round 1
Reviewer 1 Report
The manuscript is an investigation of bonding strategies within different brackets and dental materials. Surface and bracket type effects on the bond strength were investigated. The subject of the study is popular, and the text is generally well-written.
Here are my comments:
In line 86 “consisting of 44 acrylic crowns”, please provide the specific manufacturer’s information.
In line 86, the teeth were immersed in chloramine solution at 4 ºC over a period of 7 days., Could you please whether the surface change can be occurred after immersion? If happened, the investigated surface is not related to the clinical situation. Thus, I suggest investigating the surface characterization (i.e. SEM or roughness) to clarify it.
In line 102-103, please provide the related reference (publications) of Fusayama’s artificial saliva. Please further explain why you used this artificial saliva.
In Table 1 and 2, could you please explain the sample size per group is inconsistent.
In Table 2, were the shear bond strength values correlated to the tensile bond strength values?
In line 201 “roughness was higher after etching with 37 % phosphoric acid for 30 s”. The surface roughness is linked to bond strength. Please provide the surface roughness data (i.e. Ra or Sa).
In line 212 “from 70 up to 200 m”, writing error. In line 276 “study desig” ,. Please check the writing error through text.
Author Response
Dear Editor,
The manuscript entitled “Bond strength of metallic or ceramic orthodontic brackets to enamel, acrylic, or porcelain surfaces” (Manuscript ID materials-992320) has been carefully reviewed.
Please find here below answers to the reviewer(s) and editor comments:
Reviewer: 1
Comments to the Author
The manuscript is an investigation of bonding strategies within different brackets and dental materials. Surface and bracket type effects on the bond strength were investigated. The subject of the study is popular, and the text is generally well-written.
Our response: The authors acknowledge the reviewer’s comments and therefore additional information was added in the new version of the manuscript.
Here are my comments:
In line 86 “consisting of 44 acrylic crowns”, please provide the specific manufacturer’s information.
Our response: The manufacturers’ specifications have been provided in the new version of the manuscript.
Revised text: (Dentalon Plus, Kulzer GmbH, Germany);
In line 86, the teeth were immersed in chloramine solution at 4 ºC over a period of 7 days., Could you please whether the surface change can be occurred after immersion? If happened, the investigated surface is not related to the clinical situation. Thus, I suggest investigating the surface characterization (i.e. SEM or roughness) to clarify it.
Our response: The authors understand the reviewer’s concern although cleaning & storage method has been well-studied in literature regarding the removal of soft tissues maintenance of the chemical stability of the teeth tissues [31-33]. In fact, the saline solution can re-establish the hydration and balance of the mineral composition of the teeth tissues [31-33]. We cite references on the use of the storage method. Also, the authors followed the previous methods to provide a comparison of results regarding the bond strength magnitude.
In line 102-103, please provide the related reference (publications) of Fusayama’s artificial saliva. Please further explain why you used this artificial saliva.
Our response: The reference of the Fusayama’s artificial saliva solution has been provided in the new version of the manuscript. Also, we included the justification as recommended.
Revised text:The corrosive and lubricated effect of the artificial saliva solution used in this study has been reported to be similar as in human saliva [31-33].
In Table 1 and 2, could you please explain the sample size per group is inconsistent.
Our response: The number of specimens depended on the adhesion procedure and testing. Some specimens failed and therefore they were removed. However, we updated the number of specimens but still providing a consistent large size of specimens maintaining a power of statistical analysis at 100%.
In Table 2, were the shear bond strength values correlated to the tensile bond strength values?
Our response: Regarding the test set up and loading direction are quite different, we assessed the interfaces by both tests to mimic two dominant failure pathways in the oral cavity. Actually, most of studies apply only one of them to validate the bond strength results.
In line 201 “roughness was higher after etching with 37 % phosphoric acid for 30 s”. The surface roughness is linked to bond strength. Please provide the surface roughness data (i.e. Ra or Sa).
Our response: Roughness results have been provided in the new version of the manuscript.
In line 212 “from 70 up to 200 m”, writing error. In line 276 “study desig” ,. Please check the writing error through text.
Our response: The authors acknowledge for the amendments. We have performed the required corrections.
Yours sincerely,
Júlio C. M. Souza
Reviewer 2 Report
I have read with great interest the manuscript entitled “Bond strength of metallic or ceramic orthodontic brackets to enamel, acrylic, or porcelain surfaces”.
The study is interesting and was carried out with appropriate methodology and with scientific rigor, it is a interesting topic in orthodontics.
It is necessary to review the bibliographical references because they are not described according to the rules of the journal.
The authors should also summarise the conclusions of the manuscript, I believe that the conclusions are too extensive.
Author Response
Dear Editor,
The manuscript entitled “Bond strength of metallic or ceramic orthodontic brackets to enamel, acrylic, or porcelain surfaces” (Manuscript ID materials-992320) has been carefully reviewed.
Please find here below answers to the reviewer(s) and editor comments:
Reviewer: 2
Comments to the Author
I have read with great interest the manuscript entitled “Bond strength of metallic or ceramic orthodontic brackets to enamel, acrylic, or porcelain surfaces”.
The study is interesting and was carried out with appropriate methodology and with scientific rigor, it is a interesting topic in orthodontics.
Our response: The authors acknowledge the reviewer’s comments.
It is necessary to review the bibliographical references because they are not described according to the rules of the journal.
Our response: References were updated regarding the journal’ guidelines.
The authors should also summarise the conclusions of the manuscript, I believe that the conclusions are too extensive.
Our response: Conclusions have been shortened as recommended. However, we could not remove important findings regarding the extensive analysis on the bond strength of 2 different orthodontic brackets to 3 different surfaces.
Round 2
Reviewer 1 Report
The manuscript can be considered for publication.